# Evaluation of Oxidative Stress and Endothelial Dysfunction in COVID-19 Patients

**DOI:** 10.3390/medicina60071041

**Published:** 2024-06-25

**Authors:** Nurcan Kırıcı Berber, Osman Kurt, Ayşegül Altıntop Geçkil, Mehmet Erdem, Tuğba Raika Kıran, Önder Otlu, Seval Müzeyyen Ecin, Erdal İn

**Affiliations:** 1Department of Chest Diseases, Malatya Turgut Özal University, Malatya 44210, Turkey; aysegul.altintop@gmail.com; 2Department of Public Health, Faculty of Medicine, Inonu University, Malatya 44210, Turkey; osman.kurt@inonu.edu.tr; 3Department of Medical Biochemistry, Malatya Turgut Özal University, Malatya 44210, Turkey; mehmet.erdem@ozal.edu.tr (M.E.); raika.kiran@ozal.edu.tr (T.R.K.); onder.otlu@ozal.edu.tr (Ö.O.); 4Department of Occupational Medicine and Internal Medicine Clinic, Mersin City Training and Research Hospital, Mersin 33240, Turkey; seval44ecin@gmail.com; 5Department of Pulmonary Diseases, Faculty of Medicine, İzmir University of Economics, İzmir 35330, Turkey; inerda@gmail.com

**Keywords:** COVID-19, endothelial dysfunction, oxidative stress, heat shock proteins

## Abstract

*Background and Objectives*: Heat shock proteins (HSPs) are stress proteins. The endogenous nitric oxide (NO) synthase inhibitor asymmetric dimethyl arginine (ADMA) is a mediator of endothelial dysfunction. Severe acute respiratory syndrome coronavirus 2 (SARS-CoV-2) virus causes endothelial dysfunction and coagulopathy through severe inflammation and oxidative stress. Using these markers, we analyzed the prognostic value of serum ADMA and HSP-90 levels for early prediction of severe coronavirus disease (COVID-19) patients. *Materials and Methods*: A total of 76 COVID-19 patients and 35 healthy control subjects were included in this case–control study. COVID-19 patients were divided into two groups: mild and severe. *Results*: Serum ADMA and HSP-90 levels were significantly higher in the COVID-19 patients compared to the control subjects (*p* < 0.001). Additionally, serum ADMA and HSP-90 levels were determined to be higher in a statistically significant way in severe COVID-19 compared to mild COVID-19 (*p* < 0.001). Univariable logistic regression analysis revealed that ADMA and HSP-90, respectively, were independent predictors of severe disease in COVID-19 patients (ADMA (OR = 1.099, 95% CI = 1.048–1.152, *p* < 0.001) and HSP-90 (OR = 5.296, 95% CI = 1.719–16.316, *p* = 0.004)). When the cut-off value for ADMA was determined as 208.94 for the prediction of the severity of COVID-19 patients, the sensitivity was 72.9% and the specificity was 100% (AUC = 0.938, 95%CI = 0.858–0.981, *p* < 0.001). When the cut-off value for HSP-90 was determined as 12.68 for the prediction of the severity of COVID-19 patients, the sensitivity was 88.1% and the specificity was 100% (AUC = 0.975, 95% CI= 0.910–0.997, *p* < 0.001). *Conclusions*: Increased levels of Heat shock proteins-90 (HSP-90) and ADMA were positively correlated with increased endothelial damage in COVID-19 patients, suggesting that treatments focused on preventing and improving endothelial dysfunction could significantly improve the outcomes and reduce the mortality rate of COVID-19. ADMA and HSP-90 might be simple, useful, and prognostic biomarkers that can be utilized to predict patients who are at high risk of severe disease due to COVID-19.

## 1. Introduction

Severe acute respiratory syndrome coronavirus 2 (SARS-CoV-2) belongs to the Coronavirinae subfamily of the Coronaviridae family. The pneumonia epidemic caused by this virus could not be controlled and spread all over the world [1]. Human-to-human transmission is considered to occur through respiratory droplets exhaled by an infected person, so coughing and sneezing can cause SARS-CoV-2 to become airborne and allow uninfected individuals to become infected [2]. The primary site of infection of coronavirus disease (COVID-19) is the upper and lower respiratory tract. According to preliminary data from China, 81% of COVID-19 patients were reported to have mild or moderate disease, similar to the common cold and mild pneumonia, while 14% of cases were reported to have severe disease, 5% progressed to critical disease with multi-organ failure, and the mortality rate in this patient group was reported to be approximately 50% [3]. In some people with severe COVID-19, multiple organ damage and autoimmune diseases may persist for weeks, months, or even years after COVID-19 is diagnosed. As a result of these effects, people who have had COVID-19 are more likely to develop new health problems, such as diabetes, heart disease, blood clots, or strokes, compared to those who have not. This condition is called long COVID or post-COVID conditions [4].

SARS-CoV-2 primarily targets epithelial cells of the nasal, bronchial, and pulmonary epithelium through the viral structural spike (S) protein that binds to the angiotensin-converting enzyme 2 (ACE2) receptor. This receptor is expressed in the outer membranes of several other cell types—especially in vascular endothelium cells [5]. In particular, a “cytokine storm” occurs in most patients with severe disease [1]. It has been reported that a cytokine storm can induce oxidative stress through macrophage and neutrophil activation, while oxidative stress plays an important role in direct tissue damage, including mitochondrial damage in the pathogenesis of viral infection, and severe tissue damage can result in the conversion of fibrinogen into abnormal fibrin clots, leading to micro-thrombosis and pulmonary complications [6,7]. Endothelial dysfunction in COVID-19 may result in macro- and microvascular thrombotic events that may lead to impaired organ perfusion. Treatments focused on preventing and improving endothelial dysfunction can significantly improve the outcomes and reduce the mortality rate of COVID-19.

Heat shock proteins (HSPs) are stress proteins. They can be triggered when one is exposed to different types of stress [8,9,10,11]. ISPs are a large family of proteins; many different ISP molecules exist, and a large number of ISPs are expressed in cells. Hsp-90, a cellular molecular chaperone abundant in all eukaryotic cells, is also well known to play important roles in the folding process of viral capsid proteins and virion assemblies [12].

The endogenous nitric oxide (NO) synthase inhibitor asymmetric dimethyl arginine (ADMA) is a mediator of endothelial dysfunction. ADMA is a naturally occurring amino acid found in tissues and cells which circulates in plasma and is excreted in the urine [13].

Within this scope, it seems plausible that patients with pre-existing endothelial dysfunction are vulnerable to a more severe disease course, given the critical role of endothelial cells in vascular homeostasis and organ perfusion.

This study provides an overview of recent evidence linking endothelial dysfunction to COVID-19 and its potential implications for the prevention of adverse outcomes and treatment of the disease. Additionally, this study aims to determine the serum levels of ADMA and HSP-90, one of the markers of endothelial dysfunction and oxidative stress, in patients diagnosed with COVID-19, as well as to show that it can help in the early detection of individuals developing serious disease and can be used to reduce mortality in new COVID-19 outbreaks. We think that these markers can provide important clues about the pathogenesis of the disease and contribute to the literature.

## 2. Materials and Methods

### 2.1. Design of the Study and the Subjects

This case–control study included 76 consecutive patients diagnosed with COVID-19 in group 1 (mild COVID-19, 17 patients) and group 2 (severe COVID-19, 40 patients) who were hospitalized in the Pandemic Clinic between 10 June and 10 July 2021. Patients older than 18 years were included in the study, and no gender difference was observed between patients. The control group (35 healthy individuals) was age- and gender-matched with the study group.

The diagnosis of COVID-19 was based on a SARS-CoV-2-positive real-time reverse transcriptase polymerase chain reaction (RT-PCR) from a nasal and/or throat swab, combined with signs, symptoms, or radiological findings suggestive of COVID-19 infection. Clinical and laboratory parameters, as well as demographic data, of the patients were recorded. After obtaining the background and performing a physical examination, clinical evaluations of patients diagnosed with COVID-19 were performed and blood samples were collected before treatment.

Patients with COVID-19 at the time of admission were divided into 2 groups—mild and severe disease groups—according to clinical findings, respiratory rate, oxygen saturation (SpO_2_) levels, and low-dose thorax CT findings as follows.

-Mild disease: mild respiratory symptoms, positive signs of pneumonia on low-dose CT, and SpO_2_ ≥ 94% on room air.-Severe disease: any of the following:
(1)Respiratory rate > 30 breaths/min;(2)Severe respiratory distress or oxygen saturation < 90% on room air, partial oxygen saturation/fraction of inspired oxygen (PaO_2_/FiO_2_) ≤ 300 mmHg;(3)Any adult patient with COVID-19 presenting with lung infiltration > 50% on low-dose CT was defined as severe COVID-19 [14].


In the study, 35 subjects who were over 18 years of age, in the same age group as the COVID-19 cases, without any disease, and with normal physical examination were included in the control group. After clinical evaluations, including anamnesis and physical examination, were completed, blood samples were collected from the control group. Those with the following diseases were excluded from the study:

Acute myocardial infarction, acute coronary syndrome, heart failure, renal failure, chronic obstructive pulmonary disease (COPD), interstitial lung disease, any organ malignancy or immunosuppression (HIV infection, solid organ or stem cell transplantation or any immunosuppressive therapy), and pregnancy.

### 2.2. Measurement of Serum HSP-90 and ADMA Levels

Serum samples stored at −80 °C in a deep freezer were prepared for the experiment under the required conditions. HSP-90 (Cloud-Clone Corp. (Houston, TX, USA); Cat. No: SEA863Hu) and ADMA (Cloud-Clone Corp.; Cat. No: CEB301Ge) levels were determined following the manufacturer’s recommendations using commercial enzyme-linked immunosorbent assay (ELISA) kits. The absorbance of the samples was determined with a microplate reader adjusted to a 450 nm wavelength. The detection ranges of HSP-90 and ADMA were 3.12–200 ng/mL and 12.35–1000 ng/mL, respectively.

### 2.3. Statistical Analyzes

The statistical analyses were evaluated using the SPSS 22 (Statistical Package for Social Sciences; SPSS Inc., Chicago, IL, USA) package program. Descriptive data were presented as n, % values for categorical data and mean ± standard deviation values for continuous data. Chi-square analysis (Pearson Chi-square) was used to compare categorical variables between groups. The compliance of continuous variables with normal distribution was evaluated by the Kolmogorov–Smirnov test. Student’s *t*-test and the Mann–Whitney U test were used to compare paired groups, and one-way ANOVA analysis was used to compare more than two variables. The Pearson correlation test was used to examine the relationship between continuous variables. To predict the severity of COVID-19 in the study, univariate analyses were performed via binary logistic regression. In the regression model, a 95% confidence interval (CI) was employed for the calculation of the odds ratios (ORs). Within this framework, the determination of the cut-off values for ADMA and HSP-90 was performed via receiver operating characteristic (ROC) analysis. Based on the ROC curve, the performance was determined by examining the “Area under the Curve” (AUC) value. The statistical significance level was regarded as *p* < 0.05 in the analyses.

## 3. Results

### 3.1. Population Analysis

The study included 76 COVID-19 patients and 35 healthy control subjects. The mean age of the COVID-19 patients was 59.6 ± 13.4, while it was 55.5 ± 12.8 for the controls. There was no significant difference between the mean age of COVID-19 patients and control subjects (*p* = 0.133). Of the patients, 42.1% (*n* = 32) were female, while 54.3% (*n* = 19) of the control subjects were female. There was no significant difference between the COVID-19 and control groups in terms of gender (*p* = 0.232) or BMI (*p* = 0.541) (Table 1).

### 3.2. Comparison of Patients with COVID-19 and Control Groups

Platelet, lymphocyte count, lymphocyte (%), MCH (mean erythrocyte hemoglobin), plateletcrit, total protein, albumin, calcium, and sodium values were significantly lower in patients than in the control group (*p* < 0.05), whereas neutrophil count, neutrophils (%), glucose, urea, creatinine, alanine transaminase (ALT), aspartate transaminase (AST), C-reactive protein (CRP), creatine kinase (CK), creatine kinase-MB (CK-MB), gamma-glutamyl transferase (GGT), lactate dehydrogenase (LDH), neutrophil/lymphocyte ratio (NLR), platelet/lymphocyte ratio (PLR), leukocyte/albumin ratio (LAR), ADMA, and HSP-90 values were significantly higher than the control group (*p* < 0.05). Demographic and laboratory data of COVID-19 patients and the control group are shown in Table 2.

### 3.3. Comparison of Mild and Severe COVID-19 Groups

The mean age and urea of patients with severe disease were significantly higher than the mean age of patients with mild disease (*p* = 0.029). The lymphocyte percentage of patients with severe COVID-19 disease was found to be low (*p* = 0.012). On the other hand, the CRP, Pro BNP, NLR, ADMA, and HSP-90 values were significantly higher in severe COVID-19 patients (*p* < 0.05) (Table 3).

### 3.4. Evaluation of Serum ADMA and HSP-90 Levels

The mean serum ADMA levels of COVID-19 patients (mild/severe) and control subjects were 174.3 ± 18.6 pg/mL (mild), 226.2 ± 32.6 (severe) pg/mL, and 103.6 ± 11.2 (control) pg/mL, respectively. According to these results, ADMA levels were significantly higher in both mild (*p* < 0.001) and severe (*p* < 0.001) COVID-19 patients compared to control subjects (Figure 1).

The mean serum HSP-90 levels of COVID-19 patients (mild/severe) and control subjects were 10.9 ± 0.9 (mild) ng/mL, 19.2 ± 6.0 (severe) ng/mL, and 7.4 ± 1.3 (control) ng/mL, respectively. Serum HSP-90 levels were significantly higher in severe (*p* < 0.001) and mild (*p* < 0.001) COVID-19 patients compared to control subjects. Furthermore, serum HSP-90 levels were significantly higher in the severe COVID-19 group than in the mild COVID-19 group (*p* < 0.001) (Figure 1).

### 3.5. Correlation Analysis

The relationships between serum ADMA levels and various parameters were evaluated using correlation analysis. Accordingly, a positive correlation was found between ADMA and HSP-90, MCH, and urea (r = 0.577 *p* < 0.001, r = 0.257 *p* = 0.025, r = 0.474 *p* < 0.001, respectively), and a negative correlation was found between ADMA and glucose and sodium (r= −0.246 *p* = 0.032, r = −0.316 *p* = 0.005, respectively). Moreover, a positive correlation was found between serum HSP-90 and MCH and fibrinogen (r = 0.242 *p* = 0.035, r = 0.255 *p* = 0.043, respectively) (Figure 2).

### 3.6. Logistic Regression Analysis

Values with significant differences in binary comparison were subjected to binary logistic regression analysis, and according to the univariate analysis, the lymphocyte percentage (OR = 1.017, 95% CI = 1.003–1.032, *p* = 0.016), ADMA (OR = 1.099, 95% CI = 1.048–1.152, *p* < 0.001), and HSP-90 (OR = 5.296, 95% CI = 1.719–16.316, *p* = 0.004) were found to be independent variables for predicting severe disease in COVID-19 patients. When those found to be significant in the univariate analysis were included in the model for multivariate analysis, none showed significant prediction (Table 4). ADMA, HSP-90, and lymphocyte percentage had predictive value in determining severe COVID-19.

### 3.7. ROC Curve Analysis

When the cut-off value for ADMA was determined as 208.94 for the prediction of severe COVID-19 patients, the sensitivity was 72.9% and the specificity was 100% (AUC = 0.938, 95% CI = 0.858–0.981, *p* < 0.001). When the cut-off value for HSP-90 was determined as 12.68 for the prediction of severe COVID-19 patients, the sensitivity was 88.1% and the specificity was 100% (AUC = 0.975, 95% CI = 0.910–0.997, *p* < 0.001) (Figure 3).

## 4. Discussion

The pathogenesis of COVID-19 has not been fully elucidated. In particular, a “cytokine storm” occurs in most patients with severe disease [1]. The cytokine storm stimulates not only inflammation, but also immunosuppression due to apoptosis in COVID-19 [6,7,8,9,10,11,12,13,14,15]. Evidence in the literature suggests that the cytokine release syndrome is the main factor responsible for the high mortality observed in COVID-19 patients. It has been reported that the cytokine storm can induce oxidative stress through macrophage and neutrophil activation. Oxidative stress plays an important role in direct tissue damage, including mitochondrial damage in the pathogenesis of viral infection, and severe tissue damage can result in the conversion of fibrinogen into abnormal fibrin clots, leading to micro-thrombosis and pulmonary complications [6,7]. On the other hand, SARS-CoV-2 primarily targets epithelial cells of the nasal, bronchial, and pulmonary epithelium through the viral structural spike (S) protein that binds to the angiotensin-converting enzyme 2 (ACE2) receptor. This receptor is expressed in the outer membrane of several other cell types, especially in vascular endothelium cells [5]. Endothelial dysfunction is an important mechanism contributing to poor progression and adverse outcomes of COVID-19 infection. Autopsies have shown that endothelitis occurs in 90% and thrombosis in 30–60% of small and medium vessels [16]. In addition, studies involving nailfold video capillaroscopy (NVC) of the microcirculation both during and after acute infection have shown that microthrombus and flow velocity decrease in the acute period [17], while in the long term, the capillary number decreased [18] and capillary dilation occurred. It has been shown in many studies that endothelial dysfunction continues in the long term and is associated with post-COVID-19 syndrome [5,19].

Understanding the link between oxidative stress and endothelial dysfunction will lead to new therapeutic targets that will be particularly useful in severe COVID-19 cases. ADMA, an inhibitor of endogenous NO synthase, competitively inhibits endogenously produced nitric oxide synthase inhibitors (eNOS) and is, therefore, a mediator of endothelial dysfunction [20]. By inhibiting NO, ADMA decreases vascular compliance, increases vascular resistance, and limits blood flow [21]. Looking at the literature, ADMA levels are associated with various forms of pulmonary arterial hypertension (PAH) [22]. In another study, increased ADMA serum levels were associated with worse outcomes and unfavorable pulmonary hemodynamics, as well as lower survival in patients with idiopathic pulmonary arterial hypertension (IPAH) [23], and in another study, ADMA levels were significantly higher in patients with severe outcomes than in those without [24]. In a study conducted in Italy on patients with COVID-19, it was found that there was a correlation between disease severity and ADMA, and as the ADMA level increased, the need for mechanical ventilation increased [25]. However, in another study conducted in Turkey, it was found that it did not increase in the initial stages of the disease [26]. In our study, when the control group and COVID-19 patients were compared, the ADMA levels were found to be high in COVID-19 patients. On the other hand, when mild and severe COVID-19 patient groups were compared, the ADMA level was found to be high even in severe COVID-19 patients. Our study showed that the ADMA level is an important indicator of the degree and severity of the disease, following previous studies.

Hsp-90, a cellular molecular chaperone abundant in eukaryotic cells and bacteria, is also well known to play important roles in the folding process of viral capsid proteins and virion assemblies. Substrates of Hsp-90 are also involved in cell cycle regulation and signal transduction [12]. Studies have shown that HSP-90 is a host factor for influenza virus RNA polymerase [27], that it is overexpressed in damaged areas of the lungs in patients with COVID-19 [28], that HSP-90 inhibitors prevent endothelial damage and restore endothelial barrier functions [29], and that HSP-90 levels are higher in patients with severe COVID-19 [30]. In our study, serum HSP-90 levels were significantly higher in severe COVID-19 patients with more lung damage, and HSP-90 and ADMA values were found to be independent predictors of severe disease upon univariate logistic regression analysis. We think that HSP-90 and ADMA will be important markers of severity in COVID-19. NVC is an important non-invasive method for demonstrating microcirculation, and studies using NVC in combination with ADMA and HSP-90 may help to determine the severity of the disease at an early stage and to predict long COVID forms or to understand whether there are changes that may be associated with them.

When blood parameters are analyzed, lymphopenia; increases in NLR; the biochemical markers AST, ALT, and LDH; increased urea levels, increased inflammatory marker CRP; ferritin; increased erythrocyte sedimentation rates (ESRs); increased IL6 levels and TNF alpha levels; increased D-dimer levels; prolonged prothrombin times; and increased troponin levels are coagulation and cardiac markers which have prognostic values in terms of predicting mortality and disease severity [31]. Considering the literature, it was shown that inflammatory markers such as CRP, ferritin, fibrinogen, fibrinogen, and interleukin-6 can be used in the prediction of severe disease and mortality in COVID-19 [32], and in severe COVID-19 patients, it was reported that increased serum neutrophil percentages and LDH levels can be used to predict the risk of death in areas where oxygen saturation (Sa02) cannot be measured [33]. In our study, increases in glucose, urea, AST, ALT, GGT, LDH, CPR, neutrophil count, neutrophil (%), NLR, PLR, and LAR parameters were observed in COVID-19 patients compared to the control group. In addition to clinical findings, biochemical parameters contribute to the distinction between severe and mild COVID-19 infection and to the clinical decision to hospitalize patients. Thus, it was demonstrated in our study that markers can be used to differentiate patients for hospitalization and/or intensive care and to create conditions in which more opportunities are provided to patients who need it, in line with the literature.

As the inflammatory response increases, the amount of albumin in the circulation decreases; therefore, albumin is a negative acute-phase reactant [34]. In COVID-19 patients, as a result of the cytokine storm, liver damage occurs and production decreases due to this damage. Additionally, due to increased capillary permeability, albumin leaks into the interstitial space and hypoalbuminemia occurs. In various studies, low albumin levels have been shown to be a predictor of severe morbidity and mortality in COVID-19 patients [35,36,37]. In a study conducted by Rana et al., they emphasized that hypoalbuminemia is an important parameter in mortality and prognosis, that albumin levels should be closely monitored in patients, and that prognoses would be better if detected early [38]. In this study, similarly, serum albumin levels were found to be significantly low in the patient group with COVID-19. In addition, a strong negative correlation was observed between serum ADMA and HSP-90 and albumin levels.

The limitations of our study are as follows: First, our study is a single-center study, and therefore, the sample size is relatively small. Therefore, multicentric studies are needed to support our study. The second limitation of our study is that there were no patients with ARDS/critical severe pneumonia, adequate intensive care, and mechanical ventilation among our patients. Another limitation of the study is the lack of follow-up data due to the lack of long-term follow-up of the patients. Therefore, post-COVID-19 syndrome could not be evaluated in these patients in the long term.

## 5. Conclusions

As a result of our study, it was observed that ADMA and HSP-90 levels were significantly higher in COVID-19 patients, and that ADMA and HSP-90 are effective biomarkers. Further studies are needed in which serial analyses of biomarkers can be performed during hospitalization and in which the effectiveness of these two biomarkers in predicting mortality and determining the severity of the disease can be better demonstrated. Increased levels of HSP-90 and ADMA were positively correlated with increased endothelial damage in COVID-19 patients, suggesting that treatments focused on preventing and improving endothelial dysfunction could significantly improve outcomes and reduce mortality rates in COVID-19. ADMA and HSP-90 may be simple, useful, and prognostic biomarkers that can be used to determine disease severity in COVID-19 patients.

## Figures and Tables

**Figure 1 medicina-60-01041-f001:**
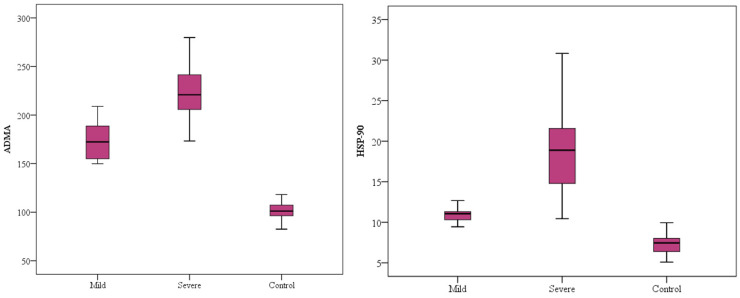
Comparison of ADMA and HSP-90 levels in all study groups.

**Figure 2 medicina-60-01041-f002:**
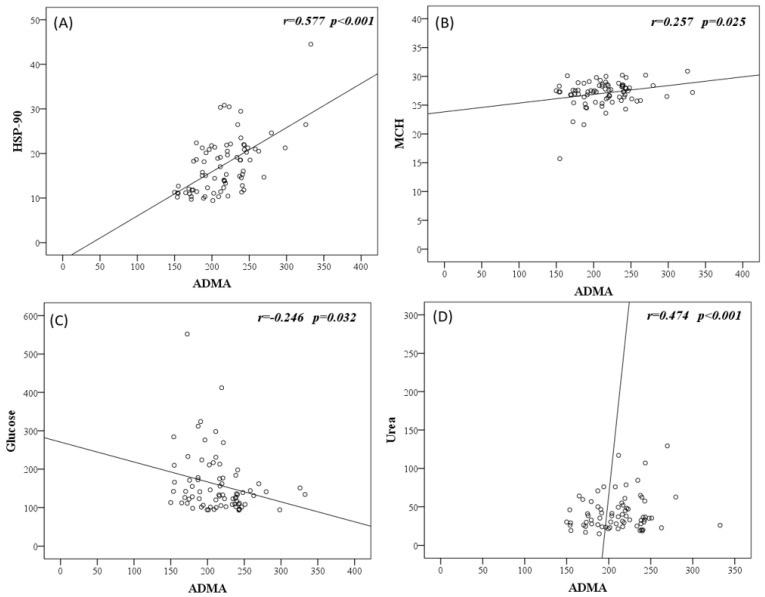
Correlation graphs of ADMA level with (**A**) HSP-90, (**B**) MCH, (**C**) glucose, (**D**) urea, and (**E**) sodium; and of HSP-90 with (**F**) MCH and (**G**) fibrinogen.

**Figure 3 medicina-60-01041-f003:**
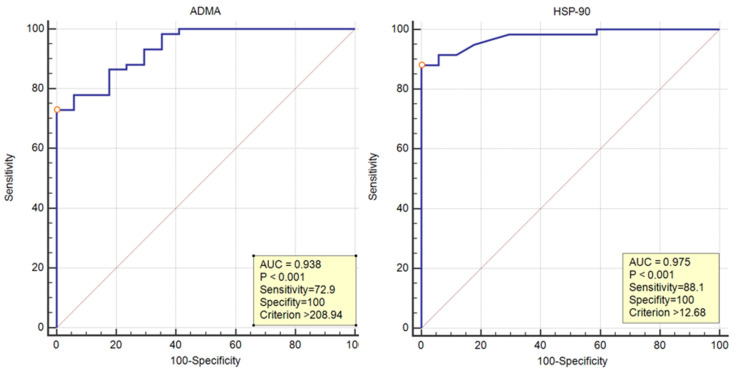
Receiver operating characteristic (ROC) curve analysis of the utility of ADMA and HSP-90 in predicting severity in COVID-19 patients.

**Table 1 medicina-60-01041-t001:** Comparison of the demographic data of patients with COVID-19 and controls.

	COVID-19 Patients (*n* = 76)	Controls (*n* = 35)	*p*
	Mean ± SD	Mean ± SD	
Age (years)	59.6 ± 13.4	55.5 ± 12.8	0.133 *
Gender (female), *n* (%)	32 (42.1)	19 (54.3)	0.232 **
BMI (kg/m^2^)	29.4 ± 5.9	30.6 ± 10.1	0.541 *

* Student’s *t*-test, ** Chi-square analysis. Abbreviations: BMI, body mass index.

**Table 2 medicina-60-01041-t002:** Comparison of laboratory data of patients with COVID-19 and controls.

	COVID-19 Patients (*n* = 76)	Controls (*n* = 35)	*p* *
	Mean ± SD	Mean ± SD	
Complete blood count			
Leukocyte (×10^9^/L)	8.2 ± 4.0	7.4 ± 1.6	0.115
Hemoglobin (g/dL)	13.2 ± 1.7	13.7 ± 1.9	0.172
Platelets (×10^9^/L)	223.8 ± 85.0	283.4 ± 60.8	**<0.001**
Neutrophil count (mcL)	6.7 ± 4.0	4.4 ± 1.3	**<0.001**
Lymphocyte count (mcL)	1.1 ± 1.0	2.6 ± 1.8	**<0.001**
Neutrophils (%)	76.7 ± 14.0	59.2 ± 8.2	**<0.001**
Lymphocytes (%)	15.8 ± 10.4	31.1 ± 7.9	**<0.001**
MCH (pg)	27.1 ± 2.2	28.0 ± 2.4	**0.048**
Plateletcrit (%)	0.2 ± 0.1	0.3 ± 0.1	**<0.001**
Biochemical markers			
Glucose (mg/dL)	159.1 ± 78.1	95.5 ± 24.6	**<0.001**
Urea (mg/dL)	43.2 ± 23.2	27.0 ± 8.5	**<0.001**
Creatinine (mg/dL)	1.0 ± 0.5	0.8 ± 0.1	**<0.001**
ALT (U/L)	42.1 ± 39.5	15.6 ± 7.9	**<0.001**
AST (U/L)	48.3 ± 34.8	16.1 ± 3.7	**<0.001**
Total protein (g/dL)	6.7 ± 0.8	7.5 ± 0.4	**<0.001**
Albumin (g/dL)	3.1 ± 0.5	4.2 ± 0.3	**<0.001**
CRP (mg/L)	7.3 ± 5.2	0.5 ± 0.3	**<0.001**
CK (U/L)	194.4 ± 181.1	89.6 ± 34.9	**<0.001**
CK-MB (U/L)	22.1 ± 13.0	10.9 ± 4.7	**<0.001**
GGT (U/L)	43.2 ± 20.4	18.1 ± 10.3	**<0.001**
LDH (U/L)	456.4 ± 239.7	183.3 ± 39.2	**<0.001**
Calcium (mg/dL)	7.9 ± 0.7	9.2 ± 0.4	**<0.001**
Sodium (mEq/L)	135.1 ± 4.2	138.3 ± 2.2	**<0.001**
Potassium (mmol)	4.6 ± 1.0	4.3 ± 0.8	0.057
NLR (%)	8.3 ± 7.1	2.0 ± 0.7	**<0.001**
PLR (%)	255.7 ± 173.4	127.5 ± 44.5	**<0.001**
LAR (%)	2.8 ± 1.6	1.8 ± 0.5	**<0.001**
ADMA (ng/mL)	214.6 ± 37.1	103.6 ± 11.2	**<0.001**
HSP-90 (ng/mL)	17.3 ± 6.4	7.4 ± 1.3	**<0.001**

* Student’s *t*-test. Note: Bold values are statistically significant values (*p* < 0.05). Abbreviations: MCH, mean erythrocyte hemoglobin; ALT, alanine transaminase; AST, aspartate transaminase; CK, creatine kinase; CK-MB, creatine kinase-MB; GGT, gamma-glutamyl transferase; LDH, lactate dehydrogenase; CRP, c-reactive protein; NLR, neutrophil/lymphocyte ratio; PLR, platelet/lymphocyte ratio; LAR, leukocyte/albumin ratio; ADMA, asymmetric dimethylarginine; HSP-90, heat shock proteins.

**Table 3 medicina-60-01041-t003:** Comparison of the demographic and laboratory data of two groups of patients with COVID-19.

	Mild (*n* = 17)	Severe (*n* = 59)	*p* *
	Mean ± SD	Mean ± SD	
Age (years)	55.0 ± 7.4	60.9 ± 14.5	**0.029**
Gender (female), *n* (%)	9 (52.9)	23 (39)	0.304 **
BMI (kg/m^2^)	28.4 ± 4.4	29.7 ± 6.3	0.426
Complete blood count			
Leukocyte (×10^9^/L)	7.3 ± 3.8	8.5 ± 4.1	0.295
Hemoglobin (g/dL)	12.6 ± 2.1	13.4 ± 1.6	0.112
Platelets (×10^9^/L)	227.5 ± 102.8	222.7 ± 80.2	0.840
Neutrophil count (mcL)	5.5 ± 3.7	7.0 ± 4.0	0.177
Lymphocyte count (mcL)	1.3 ± 0.5	1.1 ± 1.1	0.561
Neutrophils (%)	71.3 ± 12.6	78.2 ± 14.1	0.074
Lymphocytes (%)	21.3 ± 10.8	14.2 ± 9.8	**0.012**
MCH (pg)	26.2 ± 3.3	27.4 ± 1.7	0.054
Plateletcrit (%)	0.2 ± 0.1	0.2 ± 0.1	0.871
Biochemical markers			
Glucose (mg/dL)	177.9 ± 117.1	153.6 ± 63.0	0.421
Urea (mg/dL)	32.5 ± 14.2	46.3 ± 24.4	**0.029**
Creatinine (mg/dL)	0.9 ± 0.5	1.1 ± 0.4	0.341
ALT (U/L)	33.5 ± 20.8	44.5 ± 43.2	0.312
AST (U/L)	38.7 ± 23.6	51.1 ± 37.1	0.199
Total protein (g/dL)	6.9 ± 0.8	6.7 ± 0.8	0.208
Albumin (g/dL)	3.3 ± 0.5	3.0 ± 0.4	0.054
GGT (U/L)	43.2 ± 22.8	43.2 ± 19.8	0.997
LDH (U/L)	399.4 ± 254.4	472.8 ± 235.0	0.269
Calcium (mg/dL)	8.2 ± 0.4	7.9 ± 0.8	0.159
Sodium (mEq/L)	135.8 ± 4.0	134.9 ± 4.2	0.445
Potassium (mmol)	4.6 ± 0.9	4.7 ± 1.0	0.698
Inflammatory markers			
CRP (mg/L)	5.6 ± 3.0	7.8 ± 5.6	**0.040**
Procalcitonin (ng/mL), median (IQR)	0.09 (0.06–0.13)	0.13 (0.06–0.21)	0.148 ***
Ferritin (ml/ng), median (IQR)	283.8 (105.4–466.7)	445.2 (273.6–733.0)	0.059 ***
Fibrinogen (mg)	357.8 ± 116.2	403.3 ± 116.1	0.213
Cardiac marker			
Troponin I (µg/L), median (IQR)	0.27 (0.27–0.27)	0.15 (0.12–0.32)	0.769 ***
CK (U/L)	179.6 ± 188.0	198.6 ± 180.5	0.706
CK-MB (U/L)	20.7 ± 13.9	22.5 ± 12.8	0.616
NT-proBNP (pg/mL), median (IQR)	154.9 (92.7–273.2)	330.0 (152.9–598.6)	**0.028** ***
NLR (%)	5.2 ± 4.6	8.9 ± 7.0	**0.046**
PLR (%)	205.9 ± 198.8	270.0 ± 164.4	0.181
LAR (%)	2.3 ± 1.2	2.9 ± 1.7	0.126
ADMA (ng/mL)	174.3 ± 18.6	226.2 ± 32.6	**<0.001**
HSP-90 (ng/mL)	10.9 ± 0.9	19.2 ± 6.0	**<0.001**

* Student’s *t*-test, ** Chi-square analysis, *** Mann–Whitney U test. Note: Bold values are statistically significant values (*p* < 0.05). Abbreviations: MCH, mean erythrocyte hemoglobin; ALT, alanine transaminase; AST, aspartate transaminase; BMI, body mass index; CK, creatine kinase; CK-MB, creatine kinase-MB; GGT, gamma-glutamyl transferase; LDH, lactate dehydrogenase; CRP, c-reactive protein; NT-proBNP, N-terminal pro-brain natriuretic peptide; NLR, neutrophil/lymphocyte ratio; PLR, platelet/lymphocyte ratio; LAR, leukocyte/albumin ratio; ADMA, asymmetric dimethylarginine; HSP-90, heat shock proteins.

**Table 4 medicina-60-01041-t004:** Results of binary logistic regression analysis of potential predictors of mortality in COVID-19 patients.

	Univariable Model		
	B	*p*	OR
Age (years)	0.036	0.117	1.036 (0.991–1.083)
Gender (female)	0.566	0.307	1.761 (0.594–5.220)
BMI (kg/m^2^)	0.044	0.422	1.045 (0.938–1.165)
Lymphocytes (%)	−0.063	**0.020**	0.939 (0.891–0.990)
CRP (mg/L)	0.100	0.135	1.106 (0.969–1.262)
NT-proBNP (pg/mL)	0.002	0.125	1.002 (0.999–1.005)
ADMA (ng/mL)	0.094	**<0.001**	1.099 (1.048–1.152)
HSP-90 (ng/mL)	1.667	**0.004**	5.296 (1.719–16.316)

Note: Bold values are statistically significant values (*p* < 0.05). B represents the regression coefficient. OR represents the odds ratio. Abbreviations: BMI, body mass index; CRP, C-reactive protein; NT-proBNP, N-terminal pro-brain natriuretic peptide; ADMA, asymmetric dimethylarginine; HSP-90, heat shock proteins.

## Data Availability

The data that support the findings of this study are available from the corresponding author upon reasonable request.

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
