# Peer review of "Evaluation of Oxidative Stress and Endothelial Dysfunction in COVID-19 Patients"

_medicina, 2024, doi:10.3390/medicina60071041_

Round 1
Reviewer 1 Report
Comments and Suggestions for Authors
The presented research article "Evaluation of Oxidative Stress and Endothelial Dysfunction in COVID-19 Patients" is well-organized, comprehensive, and informative. Writing on COVID-19 is challenging, due to the vast and extensive literature on this subject in the last years. This paper on analyzing serum levels of ADMA and HSP-90, as potential endothelial dysfunction and oxidative stress biomarkers related to COVID-19. These biomarkers are shown to be potential indicators of disease severity, that could help to the early detection of high-risk COVID-19 patients.
Comments:
1. Abstract.
- Correct the abbreviations in your abstract (e.g. NO, SARS-CoV-2, COVID-19 etc) and grammatical errors (e.g. "35 healthy control").
- "HSP-90 (OR=5.296, 95% CI=1.719-16.316, p=0)": p=0 is there any mistake?
- "Methods section": What type of study and study design is it (cross-sectional?)? how the comparison and the matching process was conducted? what was the timing of the biomarkers that were evaluated?
2. Introduction
Please condense the introduction section. Literature for "post COVID-19 syndrome" or "long COVID-19 syndrome" should be included. How endothelial dysfunction is associated with "post COVID-19 syndrome"
3. Methods
International tools for disease severity evaluation could be used (World Health Organization-Ordinal Scale for Clinical Improvement WHO-OSCI ordinal clinical scale, COVID-19 Therapeutic Trial Synopsis 2020.) Moreover, how would you characterize the study design? (cross-sectional, case-control study?), explanations should be provided. Is the matching of sex and age enough; risk factors should be included? A one-by-one matching process could have increased the power of your study.
4. Results
Extensive population characteristics and demographics should be included in a separate table and in the Results sections respectively.
The Figures' resolution is too low. For the Tables, they need to be able to stand alone, please make sure they are clearly titled. Also, headings must be improved according to the content of each paragraph.
5. Discussion
The role of endothelial dysfunction, oxidative stress, and albumin levels in disease severity has been extensively discussed. Include more literature and clinical studies. Moreover, how endothelial dysfunction is related to post COVID-19 syndrome must be extensively discussed. The limitation section should be reworded and enriched with all the issues of the study.
Comments on the Quality of English Language
Please review for spelling and grammar. Native English language user should review the whole paper. I suggest a full proof read as a number of errors were noted, inconsistency in word capitalization etc.
Author Response
A Short Cover Letter-Reply to Comments
Reviewer 1
Commented [M1]:
- Correct the abbreviations in your abstract (e.g. NO, SARS-CoV-2, COVID-19 etc) and grammatical errors(e.g."35healthy control")."HSP-90 (OR=5.296, 95% CI=1.719-16.316, p=0)": p=0 is there any mistake?
-"Methods section": What type of study and study design is it (cross-sectional?)? how the comparison and the matching process was conducted? what was the timing of the biomarkers that were evaluated?
A1- The article error has been corrected.
-Added to the article that it is a case control study
Commented [M2]: Please condense the introduction section. Literature for "post COVID-19 syndrome" or "long COVID-19 syndrome" should be included. How endothelial dysfunction is associated with "post COVID-19 syndrome"
A2- The introduction was shortened. Suggestions were added to the article.
Commented [M3]: International tools for disease severity evaluation could be used (World Health Organization-Ordinal Scale for Clinical Improvement WHO-OSCI ordinal clinical scale, COVID-19 Therapeutic Trial Synopsis 2020.) Moreover, how would you characterize the study design? (cross-sectional, case-control study?), explanations should be provided. Is the matching of sex and age enough; risk factors should be included? A one-by-one matching process could have increased the power of your study.
A3- Changes have been made. They are indicated in the article in red.
Commented [M4]: Extensive population characteristics and demographics should be included in a separate table and in the Results sections respectively.
The Figures' resolution is too low. For the Tables, they need to be able to stand alone, please make sure they are clearly titled. Also, headings must be improved according to the content of each paragraph.
A4- The table is divided in two and added to the result section.
-Table names were also corrected after the tables were split.
-The correlation graph with low resolution was divided into two and its resolution was increased.
Commented [M5]: The role of endothelial dysfunction, oxidative stress, and albumin levels in disease severity has been extensively discussed. Include more literature and clinical studies. Moreover, how endothelial dysfunction is related to post COVID-19 syndrome must be extensively discussed. The limitation section should be reworded and enriched with all the issues of the study.
A5- Necessary suggestions were made in the discussion section and indicated in red in the article.
Commented [M6]]: Please review for spelling and grammar. Native English language user should review the whole paper. I suggest a full proof read as a number of errors were noted, inconsistency in word capitalization etc
A6- The spelling and grammar of the article in terms of the English language were re-examined.

Reviewer 2 Report
Comments and Suggestions for Authors
I read the article with great interest. The authors touched on a very relevant topic. Much of COVID-19 pathogenesis remains unexplored, and the search for diagnostic and prognostic markers for clinical diagnosis is of great importance. In addition to traditional markers of COVID-19 severity (ferritin, D-dimer, LDH and others), the authors examined the markers of endothelial dysfunction and protein stress. These indicators have a pathogenetic basis and complement the existing indicators well. The research was carried out at a good scientific level. The data that the authors collected are very interesting.
However, while reading the article, I had a few remarks about the data presented.
Remarks for Table 1.
1. In the control group, please check urea = 27.0 ±8.5. I think this is a typo.
2. In what units was total protein measured? The same in Table 2.
3. In the footnote below the Table 1 there is a transcript of NT-proBNP, but the parameter itself is not in the Table.
4. In the group of patients with COVID, some data are confusing, namely: urea 215.6±805.6, creatinine 4.1±15.8, CK 346.5±1167.7, CK-MB 27.5±51.2, GGT 57.6±65.4. Either there is an error in the data, or the distribution is not normal, in which case the data cannot be reported as Mean±SD. In this case, the data should be presented as median and interquartile range. And then the question arises about using the t-test. In this situation, you need to use the Mann-Whitney test.
5. What does double asterisk 0.232** mean (Gender, Female, %)?
Remarks according to table 2.
1. Table 2 also contains confusing data. Urea 268.3±909.1, creatinine 6.2±22.4, GGT 58.4±71.9, procalcitonin 5.4±22.0, ferritin 414.4±486.9, troponin I 0.3±-??, CK 394.6±1320.2, NT-proBNP 226.7±230.9, NT-proBNP 905.4± 1713.9, NLR 6.3±7.3, PLR 268.6±307.5. Either there is an error in the data, or the distribution is not normal, in which case the data cannot be reported as Mean±SD. In this case, the data should be presented as median and interquartile range. And then the question arises about using the t-test. In this situation, you need to use the Mann-Whitney test.
2. What does asterisk 0.304* mean (Gender, Female, %)?
Author Response
Reviewer 2
Commented [M1]]
Remarks for Table 1.
- In the control group, please check urea = 27.0 ±8.5. I think this is a typo.
A1- The urea value has been corrected. it is indicated in red color in the article.
- In what units was total protein measured? The same in Table 2
A2- The measured total protein unit is shown in red in the article. Corrected in table 2 and indicated in red color.
- In the footnote below the Table 1 there is a transcript of NT-proBNP, but the parameter itself is not in the Table.
A3- The abbreviation in the footnote has been removed.
- In the group of patients with COVID, some data are confusing, namely: urea 215.6±805.6, creatinine 4.1±15.8, CK 346.5±1167.7, CK-MB 27.5±51.2, GGT 57.6±65.4. Either there is an error in the data, or the distribution is not normal, in which case the data cannot be reported as Mean±SD. In this case, the data should be presented as median and interquartile range. And then the question arises about using the t-test. In this situation, you need to use the Mann-Whitney test.
A4- Necessary corrections made.
- What does double asterisk 0.232** mean (Gender, Female, %)?
A5- In that table, one star indicates Student's t-test and two stars indicate chi-square analysis.
Commented [M2]]
Remarks according to table 2.
- Table 2 also contains confusing data. Urea 268.3±909.1, creatinine 6.2±22.4, GGT 58.4±71.9, procalcitonin 5.4±22.0, ferritin 414.4±486.9, troponin I 0.3±-??, CK 394.6±1320.2, NT-proBNP 226.7±230.9, NT-proBNP 905.4± 1713.9, NLR 6.3±7.3, PLR 268.6±307.5. Either there is an error in the data, or the distribution is not normal, in which case the data cannot be reported as Mean±SD. In this case, the data should be presented as median and interquartile range. And then the question arises about using the t-test. In this situation, you need to use the Mann-Whitney test.
A1- Necessary corrections made.
- What does asterisk 0.304* mean (Gender, Female, %)?
A2- In that table, one star indicates Student's t-test and two stars indicate chi-square analysis.

Reviewer 3 Report
Comments and Suggestions for Authors
The paper is quite well written. The article covers an interesting and current topic. Nevertheless, in my opinion, some parts need to be improved, I have some comments:
1) Abstract. Conclusion: ADMA and HSP-90 might be simple, useful, and prognostic biomarkers that can be utilized to predict patients who are at high risk of severe disease in COVID-19 patients. Abstract might be beneficial to include a sentence that briefly summarizes the key findings of the study. This can provide readers with a quick overview of the research.
2) The primary site of infection of COVID-19 is the upper and lower respiratory tract. According to preliminary data from China, 81% of COVID-19 patients were reported to have mild or moderate disease, similar to the common cold and mild pneumonia, while 14% of cases were reported to have severe disease, 5% progressed to critical disease with multi-organ failure, and the mortality rate in this patient group was reported to be approximately 50%.[7] Early recognition of severe forms of the disease is crucial to determine treatment strategies and early hospitalization and treatment of at-risk patients. Patients' clinical status, oxygen saturation, and comorbidities largely determine the need for hospitalization, while some laboratory parameters can facilitate the assessment of disease severity. [8]
3) . In light of the potential association of COVID-19 with endothelial damage, it seems plausible that patients with pre-existing endothelial dysfunction are vulnerable to a more severe disease course, given the critical role of endothelial cells for vascular homeostasis and organ perfusion.
4) This study provides an overview of recent evidence linking endothelial dysfunction to COVID-19 and its potential implications for the prevention of adverse outcomes and treatment of the disease. This study aims to analyze markers of endothelial dysfunction and oxidative stress in patients diagnosed with COVID-19 and to show that serum levels of ADMA and HSP-90 markers can help in the early detection of individuals who develop severe disease and can be used to reduce mortality in new COVID-19 outbreaks. ... Please improve the description of study aim and explain all the acronyms.
5) 2. Materials and Methods 2.1. Study Design and Subjects This study included 57 consecutive patients diagnosed with COVID-19 in group 1 (mild COVID-19, 17 patients) and group 2 (severe COVID-19, 40 patients) who were hospitalized in the Pandemic Clinic of Malatya Training and Research Hospital in Turkey between June 10, 2021, and July 10, 2021, and met the inclusion criteria. Please, underline the inclusion and the exclusion criteria.
6) 3. Results. Underline the most important results to clarify the data.
7) 4. Discussion In our study, where we analyzed the efficacy of two new biomarkers, ADMA and HSP-90, which can help in the early detection of individuals who may develop severe disease in hospitalized COVID-19 patients and can be used to reduce mortality in new COVID-19 outbreaks, serum ADMA and HSP-90 levels were found to be higher in COVID-19 patients compared to healthy controls. In addition, when severe COVID-19 patients were compared with mild COVID-19 patients, serum ADMA and HSP levels were higher in severe COVID-19 patients. The discussion section needs to be improved. It is necessary to clarify the results obtained and compare them with previous or similar articles published in literature.
Comments on the Quality of English LanguageModerate changes of English language are required
Author Response
Reviewer 3
Commented [M1]]
-Abstract. Conclusion: ADMA and HSP-90 might be simple, useful, and prognostic biomarkers that can be utilized to predict patients who are at high risk of severe disease in COVID-19 patients. Abstract might be beneficial to include a sentence that briefly summarizes the key findings of the study. This can provide readers with a quick overview of the research.
A1- Your suggestions have been made. they are indicated in the article in red color.
Commented [M2]
-The primary site of infection of COVID-19 is the upper and lower respiratory tract. According to preliminary data from China, 81% of COVID-19 patients were reported to have mild or moderate disease, similar to the common cold and mild pneumonia, while 14% of cases were reported to have severe disease, 5% progressed to critical disease with multi-organ failure, and the mortality rate in this patient group was reported to be approximately 50%.[7] Early recognition of severe forms of the disease is crucial to determine treatment strategies and early hospitalization and treatment of at-risk patients. Patients' clinical status, oxygen saturation, and comorbidities largely determine the need for hospitalization, while some laboratory parameters can facilitate the assessment of disease severity. [8]
A1- Your suggestions have been corrected in the article.
Commented [M3]
-In light of the potential association of COVID-19 with endothelial damage, it seems plausible that patients with pre-existing endothelial dysfunction are vulnerable to a more severe disease course, given the critical role of endothelial cells for vascular homeostasis and organ perfusion.
A3- Ä°ndicated in red color in the article.
Commented [M4]
-This study provides an overview of recent evidence linking endothelial dysfunction to COVID-19 and its potential implications for the prevention of adverse outcomes and treatment of the disease. This study aims to analyze markers of endothelial dysfunction and oxidative stress in patients diagnosed with COVID-19 and to show that serum levels of ADMA and HSP-90 markers can help in the early detection of individuals who develop severe disease and can be used to reduce mortality in new COVID-19 outbreaks. ... Please improve the description of study aim and explain all the acronyms.
A4- Ä°ndicated in red color in the article.
Commented [M5]
-2. Materials and Methods 2.1. Study Design and Subjects This study included 57 consecutive patients diagnosed with COVID-19 in group 1 (mild COVID-19, 17 patients) and group 2 (severe COVID-19, 40 patients) who were hospitalized in the Pandemic Clinic of Malatya Training and Research Hospital in Turkey between June 10, 2021, and July 10, 2021, and met the inclusion criteria. Please, underline the inclusion and the exclusion criteria.
A5- Ä°ndicated in red color in the article.
Commented [M6]
-3. Results. Underline the most important results to clarify the data.
A6-- Ä°ndicated in red color in the article.
Commented [M7]
-4. Discussion In our study, where we analyzed the efficacy of two new biomarkers, ADMA and HSP-90, which can help in the early detection of individuals who may develop severe disease in hospitalized COVID-19 patients and can be used to reduce mortality in new COVID-19 outbreaks, serum ADMA and HSP-90 levels were found to be higher in COVID-19 patients compared to healthy controls. In addition, when severe COVID-19 patients were compared with mild COVID-19 patients, serum ADMA and HSP levels were higher in severe COVID-19 patients. The discussion section needs to be improved. It is necessary to clarify the results obtained and compare them with previous or similar articles published in literature.
A7-- At the bottom I deleted this information because the markers are compared with the literature.

Round 2
Reviewer 1 Report
Comments and Suggestions for Authors
All my previous comments and recommendations have been thoroughly addressed by the authors.
Comments on the Quality of English LanguageAll my previous comments and recommendations have been thoroughly addressed by the authors.
Author Response
Commented [M1]:
- Minor editing of English language required
A1- English language has been revised.

Reviewer 3 Report
Comments and Suggestions for Authors
The manuscript has been improved. Finally, I suggest to add some references on endotelial damage in COVID-19. I suggest some articles such as:
- Detailed videocapillaroscopic microvascular changes detectable in adult COVID-19 survivors. Microvasc Res. 2022;142:104361. doi:10.1016/j.mvr.2022.104361
- Microvascular Alteration in COVID-19 Documented by Nailfold Capillaroscopy. Diagnostics (Basel). 2023;13(11):1905. Published 2023 May 29. doi:10.3390/diagnostics13111905
- Nailfold capillaroscopy findings in patients with coronavirus disease 2019: Broadening the spectrum of COVID-19 microvascular involvement. Microvasc Res. 2021;133:104071. doi:10.1016/j.mvr.2020.104071
Author Response
Commented [M1]]
The manuscript has been improved. Finally, I suggest to add some references on endotelial damage in COVID-19. I suggest some articles such as:
- Detailed videocapillaroscopic microvascular changes detectable in adult COVID-19 survivors. Microvasc Res. 2022;142:104361. doi:10.1016/j.mvr.2022.104361
- Microvascular Alteration in COVID-19 Documented by Nailfold Capillaroscopy. Diagnostics (Basel). 2023;13(11):1905. Published 2023 May 29. doi:10.3390/diagnostics13111905
- Nailfold capillaroscopy findings in patients with coronavirus disease 2019: Broadening the spectrum of COVID-19 microvascular involvement. Microvasc Res. 2021;133:104071. doi:10.1016/j.mvr.2020.104071
A1- Some suggested references about endothelial damage in COVID-19 were read and added to the article, and this is indicated in red in the article.( 275 and 279 specified in line and 306 and 309 specified in line). Finally, 19,20,21 were added as references.
